# Twitter (X) use predicts substantial changes in well-being, polarization, sense of belonging, and outrage
Victoria Oldemburgo de Mello [1] ✉, Felix Cheung [1,3] & Michael Inzlicht [2,3]

In public debate, Twitter (now X) is often said to cause detrimental effects on users and society. Here we address this research question by querying 252 participants from a representative sample of U.S. Twitter users 5 times per day over 7 days (6,218 observations). Results revealed that Twitter use is related to decreases in well-being, and increases in political polarization, outrage, and sense of belonging over the course of the following 30 minutes. Effect sizes were comparable to the effect of social interactions on well-being. These effects remained consistent even when accounting for demographic and personality traits. Different inferred uses of Twitter were linked to different outcomes: passive usage was associated with lower well-being, social usage with a higher sense of belonging, and information-seeking usage with increased outrage and most effects were driven by within-person changes.

Social media has become so widespread[1] that even if it had only a small impact on people, it could produce meaningful changes in society. It is thus critical to evaluate how and if social media impacts its users. Here, we delve into the daily experience of social media users by focusing on one particularly influential social media platform, Twitter.

Although small in comparison to other platforms such as Instagram and Tik Tok, Twitter (now X) has received outsized attention because it hosts elites in entertainment, journalism, and politics. As such, the experience on Twitter has the potential to shape how elites (and by consequence, the public) regard, depict, and regulate social media. Despite its importance, there have been few systematic attempts to describe the impact of using Twitter. Most Twitter research examines how public data, such as tweets and followings, relate to political polarization[2,3], expressions of outrage[4,5], and subjective well-being[6]. Nevertheless, public Twitter data does not reliably represent users' thoughts: 80% of the tweets on the platform were produced by 10% of the users, who tend to be more engaged with and tweet more about politics than the average user, who tends to be younger, richer, more educated, and more engaged with politics than the average U.S. adult[7,8]. Thus, it remains unclear whether these findings generalize to most social media users or even most Twitter users.

As platforms constantly introduce changes and get replaced, research on their effects can easily become obsolete. To avoid obsolescence, we examine the psychological impact of Twitter use as a function of its features and affordances, thereby facilitating the translation of findings to different platforms and time periods. In addition, we also explore potentially

beneficial effects of social media use (such as sense of community). Towards these ends, we examine the prevalence of the different forms of Twitter use and how they relate to subjective well-being, political polarization, outrage, and sense of belonging in a sample of users more representative of the population than previous studies. Please note, we refer to the name of the platform as Twitter, not X, as the data we obtained before the platform changed names. Where applicable, we will generalize the results to X.

How does Twitter use relate to subjective well-being? Although the effects of Twitter use on subjective well-being—defined as how individuals evaluate their life satisfaction and feel about their own lives, including both positive and negative emotions[9]—have not been thoroughly investigated, research on social media more generally abounds.

In the past few years, studies suggesting that social media is to blame for a recent worsening of subjective well-being and related constructs (such as depression and loneliness)[10–15] gained media attention[16]. Nevertheless, these effects are contested. Some researchers argue that the effect sizes of social media use on well-being are so small that they have no practical significance[17,18]; others claim that the relationship is likely nonexistent[19–22], and that significant effects are explained by confounding variables, such as personality[23,24]. This debate has led to the development of a line of research focused on fundamental questions in social media research: can self-reported measures of social media use—used in most research and only modestly correlated with actual social media use[25,26]—be trusted? Can we be sure about the causal links between social media use and well-being without disentangling the processes that happen intra-individually (within a person

[1]University of Toronto, Toronto, ON, Canada. [2]Rotman School of Management, Toronto, ON, Canada. [3]These authors jointly supervised this work: Felix Cheung, Michael Inzlicht. ✉e-mail: victoria.mello@mail.utoronto.ca

over time) from those that happen between people[27]? Despite the lack of consensus, we address these issues by using more ecologically valid measures of Twitter use and statistical analyses that allow the disentangling of effects at different levels of analysis.

Affective polarization is the emotional component of political polarization. It is the degree to which people dislike and distrust the other side of the political spectrum[28]. How does it relate to Twitter use? Most works on the subject examine the connectivity among users (through followings, replies, and retweets) to investigate the extent to which networks are segregated[3,29]. Although these studies provide some evidence that Twitter networks tend to be politically segregated, they often overlook whether these structures relate to actual feelings and attitudes towards political issues.

Studies done on other social media platforms have examined affective components of polarization. In two studies, U.S. participants who were randomly assigned to leave Facebook for a week or a month were less polarized than participants who remained on Facebook as usual[30,31]. However, a replication of this experiment in a sample comprised of Bosnians, Serbs, and Croats found opposite effects[32], suggesting that the relationship between social media use and polarization is not so straightforward. These inconsistent findings suggest that the relationship might be moderated by other variables, such as the degree to which one's online network is segregated in comparison to one's offline network. Besides, only a small minority of users engage with political content on social media; most people use social media to keep in touch with family and friends[33]. Political engagement on social media platforms seems to be more prevalent among intellectual elites, many of whom are found on Twitter but whose experience is not representative of the larger population[34].

Outrage is a class of emotions (usually anger, contempt, and disgust) that follow moral judgments[35]. Although relatively rare in life, expressions of outrage seem to be more prevalent on social media[5] Research suggests that this phenomenon is driven by the algorithmic amplification of outrage—as a highly arousing emotion, users are more likely to engage with content that elicits outrage *and* produce content that provokes outrage in others[36]. For instance, each moral-emotional word in a politics-related tweet increases the chances of a retweet by 20%[4]. Yet, it remains unclear whether online expressions of outrage are merely performative (to generate engagement) or if they originate from real high-arousing emotions.

Establishing and maintaining relationships is a human need. As such, social relationships are predictors of psychological health[37]. Social media platforms emerged as an online alternative to in-person socialization, but can they satisfy the need for social connection? Research dating back to the early years of social media linked social media presence with increases in social capital, meaning that people who used social media tended to have more friends and a better social life[38,39]. Nevertheless, not much attention has been given to this topic recently. Thus, we aim to understand how Twitter use might contribute to this important but neglected mental health predictor, conceptualizing it as a sense of belonging—the subjective experience of being valued and respected as part of a group.

Even if social media had a clear net negative impact on people, it is possible that some aspects of social media are beneficial to some while other aspects are detrimental to others: a more nuanced approach would allow us to better maximize its benefits while minimizing its harms[24]. Thus, to understand the psychological effects of social media use it is critical to consider the contextual variables that might moderate the relationship: who uses social media, how they use it, and why they use it.

Social media use, as a rich and complex behavior, can be categorized in numerous ways. Previous studies have attempted to classify social media use as passive or active, yielding promising results[40,41]. However, recent reviews indicated inconsistent outcomes due to the absence of standardized operationalization[42]. To achieve a comprehensive categorization of social media use, it is crucial to analyze both the structural and functional aspects of these behaviors. The structural approach entails identifying the affordances of social media platforms, such as messaging, liking, and scrolling through a feed. In contrast, the functional approach involves examining the purpose of these actions, such as scrolling the feed to alleviate boredom or search for news. This approach facilitates the examination of reasons for moderating effects (e.g., social media seems to be more detrimental to girls, but why?)[15] Characteristics of the user might also be important moderators for the effects of social media use. Personality traits, for instance, are linked to different types of social media usage: Neuroticism, Agreeableness, and Extraversion are positively related with self-expression on social media[43].

Here, we investigate everyday use of Twitter and estimate its effects on subjective well-being, experienced emotions, political polarization, outrage, and sense of belonging. Critically, we move beyond simply probing the average effect of social media use[24] and instead ask how these effects (if any) are moderated by *who* the person is (e.g., personality traits, age, gender), *what* they are doing on social media (e.g., commenting, passive scrolling, etc.), and *why* they are using social media (e.g., for entertainment, news consumption, etc.).

To address the limitations of prior social media studies (e.g., cross-sectional methods and retrospective self-reports)[24,44], we conduct an experience sampling study. This method involves surveying participants multiple times a day over several days to capture the occurrence of events and within-person changes over time. Although still a self-report method, the experience sampling approach attempts to mitigate some of the common challenges associated with self-report methods. Traditional self-report methods often face issues such as recall bias, where participants might not accurately remember past events. Experience sampling offers high ecological validity and low recall biases[45], complementing existing social media research methods. It also enables the separation of between and within-subject effects and the examination of causal heterogeneity[46]. Given that within groups associations do not translate into within individual associations, processes at both levels of analysis should be examined[27].

## Method
### Sampling strategy
Using multi-level simulations[47] (with $d = 0.15$ and 80% statistical power), we determined that a sample size of 220–300 participants (each providing 20 observations) would suffice. We ultimately aimed to collect data from 300 participants. To get as close as possible to a representative sample of Twitter users in age, gender, and race, we used quota sampling on Prolific Academic[48], where we screened participants for Twitter use, inviting those who used the platform at least twice a week to participate in our study. 404 participants accepted the invitation to join the study and answered the baseline survey. We ran the screening and the invitations to the study in batches (8 in total) until we had at least 300 participants answering at least one of the experience sampling surveys ($N = 309$). Data was collected from March to June 2021. This study was approved by the Research Ethics Board at the University of Toronto. All participants gave informed consent before joining the study.

After excluding participants for failing attention checks or not responding to enough surveys (less than 9), our final sample was reduced to 252 participants. A sensitivity analysis revealed that our final sample still allowed to detect effect sizes $d = 0.15$ for within-subject effects and $d = 0.35$ for between-subject effects considering 80% power. These individuals had an average age of 42.99 years (SD = 14.06), were 51% male, and had a racial composition roughly representative of the US population (72% white, 14% black, 7.5% Asian, 4% mixed race, and 2.5% other ethnicities). 67.5% of the sample self-identified as liberal, 13.5% as conservative, 16.3% as moderates, and 2.8% did not report political ideology. On Twitter, 14% of the users consider themselves to be very conservative and about 64% of heavy users identify as Democrats (compared with 55% of light users)[7]. These numbers suggest that our sample was roughly representative of Twitter user in terms of political ideology. While our sample was not strictly representative, it was an improvement over many social media studies that rely on convenience samples of undergraduate students or focus on the most active Twitter users (Pew Research Center, 2021).

## Procedure

After participants joined the study on Prolific, they completed a baseline survey that included the Big Five personality questionnaire[49] and more information about the experience sampling study. We included but did not analyze numerous other variables. For a full list of the measures see the supplementary methods in the Supporting Information file. Figure 1 contains a flowchart of the study procedures.

After participants answered the baseline survey, they were invited to join the experience sampling study. To get a representative sample of people's experience, the experience sampling surveys were sent five times a day for seven days between 9 am and 10 pm at random times with an interval of at least two hours between surveys; they answered a total of 6218 surveys (see detailed study procedures on Fig. 1). The surveys were sent via text message with SurveySignal and distributed via Qualtrics. Participants had up to 40 min to answer the survey before the link expired. In these surveys, participants were asked if they had used Twitter in the past 30 min (yes/no); and if they answered "yes", we explored further details of their experience.

Surveys asked participants if they had used Twitter in the past 30 min; if they said yes, they were probed to provide details of their usage. If they said no, they completed filler questions so that the survey would be the same length had they reported using Twitter or not. To examine the different types of use, participants next reported on their Twitter behaviors (*what* they were doing on Twitter, for example passively scrolling or commenting) and Twitter functions (*why* they were using Twitter, for example to seek information or for entertainment). Participants were then asked to report on their current levels of well-being, experienced emotion, affective polarization, and sense of belonging.

We conceptualized Twitter behaviors as observable actions or activities they could do on Twitter, such as tweeting or messaging. To obtain that information, we asked participants to select which behaviors they had performed on Twitter in the past 30 min from a list of behaviors containing (1) scrolling down the feed, (2) liking tweets, (3) retweeting tweets, (4) tweeting, (5) making a comment, (6) messaging, (7) seeing the trending topics, and (8) seeing others' profile. Participants indicated all the behaviors they performed in binary form. We also asked participants if they had interacted "with anyone who holds different values or a different worldview" from theirs on Twitter ("yes" or "no").

We conceptualized Twitter functions (the *why* of Twitter use) as the underlying motivations or reasons for the Twitter behaviors. For instance, I can comment on a tweet because I want to socialize with others or because I want to relieve my boredom. To get a sense of the possible functions, we consulted qualitative research on why people use social media[50] and then created and piloted a scale that captures functions of social media use into five factors: using social media for entertainment, escapism, social interaction, self-promotion, and information seeking. All details of the development of the *Functions of Social Media Use* scale can be found under supplementary methods and tables S1 and S2 in the Supporting Information file. In the experience sampling surveys, participants answered "why did you use Twitter in the past 30 min?" by selecting one or more options from a list containing "To entertain myself/have fun." (*entertainment* factor), "To distract myself from stressful events." (*escapism* factor), "To interact with a community or group of people." (*social interaction* factor), "To promote myself or my work." (self-promotion factor), "To seek information or inspiration." (*information* seeking factor). All responses were recorded in binary form. Participants who indicated that they had not used twitter in the past 30 min were provided with filler questions so that the survey would have the same length had they reported using Twitter or not.

Finally, we measured our dependent variables. First, we asked participants about their current levels of well-being with a modified version of the Scale of Positive and Negative Experience[51], a scale that asks participants to indicate the extent to which they have felt 13 different emotions in the past 30 min. We modified the scale to add some emotional experiences we were interested in (bored, lonely, anxious, disgusted, repulsed, excited, tired) and excluded some items that were somehow redundant (good, bad, pleasant, unpleasant, happy, contented) to keep it short. From the original scale, we kept *positive, negative, sad, afraid, joyful,* and *angry*. Participants indicated in a 5-point Likert scale the extent to which they have felt each emotion ranging from "none at all" to "a great deal". We conceptualized momentary well-being as the difference between the mean of positive emotions (positive and joyful) and the mean of negative emotions (negative, sad, afraid, and angry). We conceptualized outrage as the sum of the emotions *angry, disgusted,* and *repulsed* (Cronbach's alpha of 0.80).

We then prompted participants about their sense of belonging in an online community with a two-item measure adapted from the sense of belonging in sport scale[52] in which participants answered in a 5-point Likert scale ranging from "Strongly Agree" to "Strongly Disagree". The items asked participants the extent to which they felt like they belonged to a community and the extent to which they felt valued and respected in a community. To inquire about participants' current levels of affective polarization, we used a thermometer scale that asked them to rate from 0 to 100 how warm they felt towards Democrats and Republicans[53]. Polarization was operationalized as the absolute value of the difference between warmth towards Democrats and Republicans. We also wanted to know the effects of social interactions on well-being, so we asked participants if they had any social interaction in the past 30 min, if that interaction was online or in person, and the quality of that interaction in a 5-point Likert scale ranging from "very negative" to "very positive".

Participants were compensated for their participation according to how much data they provided. All participants who answered the baseline survey and at least one of the experience sampling surveys received $5. Participants could earn up to $25 if they answered more than 90% of the surveys and proved that they had an active Twitter account (by providing a Twitter handle of an existing account).

Participants have filled out, on average, 24.67 surveys out of the 35 sent (calculated across participants), with an SD of 7.15. The median of responses across participants was 26, and the mode was both 31 and 32 (17

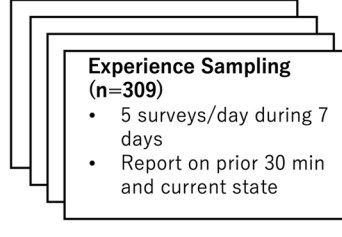

**Fig. 1 | Flowchart of study procedures.** Participants were pre-screened based on Twitter use frequency (left box). Participants then answered the baseline survey (center box) and were invited to join the experience sampling study (right box), where they received 5 surveys a day for 7 days asking about their social media use and psychological outcomes.

observations each). The lowest number of surveys that a participant could answer and still be included in our analysis was 9 (the actual sample minimum). 11 participants answered all 35 surveys.

## Statistical analysis

We used multilevel Bayesian models using the package brms[54] in R 4.0.3[55] to analyze the data. We were interested in the average effects and in the variability of the effect among participants, so we used models with random intercepts and random slopes. The outcome variables were z-scored to facilitate comparison of the magnitude of effects among different DVs. To prevent the confounding of different levels of analysis, we divided our predictors into between-subjects means and within-subjects deviations[56]. This allowed us to differentiate when the effects were driven by within-person differences or between-person differences. To estimate the practical significance of our effects, we compared coefficients with a positive control, a benchmark predictor that had a predictable and meaningful effect on one of the DVs.

An example of the model is in Eq. (1), where Y is the outcome of interest, X is the predictor, and t is the number of the observation (such that t - 1 is a lagged observation). In all models, we controlled for the lagged outcome variable ($Y_{t-1}$) to remove the variance explained by previous levels of the outcome variable. We also controlled for past levels of the predictor ($X_{t-1}$) to remove the variance that might be related to previous experiences. These controls allowed us to estimate the association between Twitter use and changes in the outcome variable of interest (e.g., the difference in well-being between two time points a few hours apart).

$$Y = Y_{t-1} + X + X_{t-1} + (Y_{t-1} + X + X_{t-1} \mid participant) \quad (1)$$

For each model, we used 4 Markov chains Monte Carlo with 10,000 iterations each, discarding the first 5000. The total number of draws after warmup was 20,000. We did not specify any priors, so the models used the brms default (flat). For each result, we report the mean of the posterior samples as well as two-sided 95% Credible Intervals (lower 95% CI and upper 95% CI). Data, materials, and code can be found at osf.io/e8krz.

## Preregistration

We initially preregistered our study protocol at osf.io/en92q. However, during our research, the data we collected prompted a significant shift in our analysis. Instead of testing the preregistered hypotheses, then, we adapted our approach to explore the correlations of Twitter use more thoroughly. We also used robust Bayesian analyses, something we did not foresee. This major deviation from our original plan, driven by the nature of the rich data we collected, means that the current study diverged significantly from its preregistration—so we no longer consider it a preregistered study. However, for the sake of transparency, we note and link to the original preregistration and justified our decision to not consider this study preregistered.

## Results

To analyze the data, we used Bayesian multilevel models controlling for past levels of the dependent variable and the independent variable—these controls allowed us to better isolate effects of Twitter use in the past 30 min on recent changes in the outcome variables. To avoid conflating cross-sectional and longitudinal effects, all our predictors were separated into two variables: one containing the variance of individual differences (the participant's mean for that variable, labeled as *between-person level* of the predictor) and another containing the variance of participants' change over time (score mean-centered around participant's mean, labeled *within-person level*). This strategy allowed us to disentangle effects at different levels. All outcome variables were z-scored to facilitate comparison. We considered associations as substantial if they did not include zero in their 95% credible intervals. We report point estimates and 95% credible intervals to facilitate interpretation.

## Description of everyday Twitter use

On average, participants reported being on Twitter on 26.7% of the surveys they answered (SD = 21%). Most of the Twitter use was passive, as active Twitter behaviors (Tweeting or Retweeting) were reported on average in only 18.2% of the surveys in which they used Twitter (SD = 29.4%). The most common behavior on Twitter was *scrolling down the feed*; participants reported this behavior in 74% of the surveys they reported Twitter use. The least common behavior was messaging other people on Twitter, reported in 7% of the survey where Twitter use was reported. A breakdown of all Twitter behaviors is on Fig. 2. The co-occurrence of behaviors and functions can be seen in Fig. 3.

We also asked participants why they used Twitter in the past 30 min and let them select as many options as they wanted from the *functions of social media use* scale. Participants reported having used social media for "entertainment" purposes most of the time (66% of the surveys), followed by "information seeking" (49%), "interacting with others" (23%), "escapism" (18%), and "self-promotion" (2%). Participants reported encountering people whose opinions differed from theirs 14% of the time they were on Twitter (SD = 24%).

## Twitter use, well-being, and emotions in daily life

To assess well-being, participants answered a modified version of the Scale of Positive and Negative Experience[51], indicating the extent to which they have felt 13 different emotions in the past 30 min. Momentary well-being was conceptualized as the difference between the mean of positive emotions (*positive* and *joyful*) and the mean of negative emotions (*negative*, *sad*, *afraid*, and *angry*).

When participants used Twitter, they reported a decrease of .10 SD in well-being, $b_{within}$ = 0.10, CrI = [−0.15, −0.04] (Fig. 4). That effect was found only at the within-person level. To estimate how meaningful the effects were, we used a benchmark from our own sample. In our survey, we asked participants if they had personally interacted with others in the past 30 min. Because social interactions have a meaningful and overall positive effect on people's well-being[57], estimating its effect on well-being in our sample serves as a good benchmark to compare against our effect sizes.

We found that having an in-person social interaction in the past 30 min (versus not) was related to a 0.15 SD increases in well-being (within-person), $b_{within}$ = 0.15, CrI = [0.02, 0.28]. The coefficient for the effect of Twitter use on well-being ($b_{within}$ = −0.10) was in the opposite direction but had two-thirds of the magnitude of our benchmark, suggesting substantial effects.

We also investigated the effects of Twitter use on reported boredom, anxiety, and loneliness, emotions that could complement the relationship with well-being (all measured as individual items in the Scale of Positive and Negative Experience). Twitter use was related to a 0.22 SD increase in boredom at the within-person level, $b_{within}$ = 0.22, CrI = [0.14, 0.30], and a 0.50 SD increase at the between person level, $b_{between}$ = 0.50, CrI = [0.18, 0.83]. Both coefficients were larger than our benchmark, again suggesting substantial effects (Fig. 4). Twitter use was related to a 0.42 SD increase in loneliness, only at the between-person level (i.e., people who use Twitter more are, on average, lonelier), $b_{within}$ = 0.42, CrI = [0.05, 0.78]. Twitter use was not related to anxiety at any level of analysis, $b_{within}$ = 0.02, 95% CrI = [−0.03, 0.08], $b_{between}$ = 0.16, 95% CrI = [−0.18, 0.50].

## Twitter use and sense of belonging

Sense of belonging online was measured using a two-item scale that asked participants to express the extent to which they felt like they belonged to a community and the extent to which they felt valued and respected in a community[52] in a 5-point Likert scale ranging from "Strongly Agree" to "Strongly Disagree".

Twitter use was related to a 0.11 SD increase in sense of belonging, $b_{within}$ = 0.11, CrI = [0.04, 0.17], only at the within-person level with a coefficient close to the benchmark, suggesting a meaningful association between Twitter use and sense of belonging (Fig. 4).

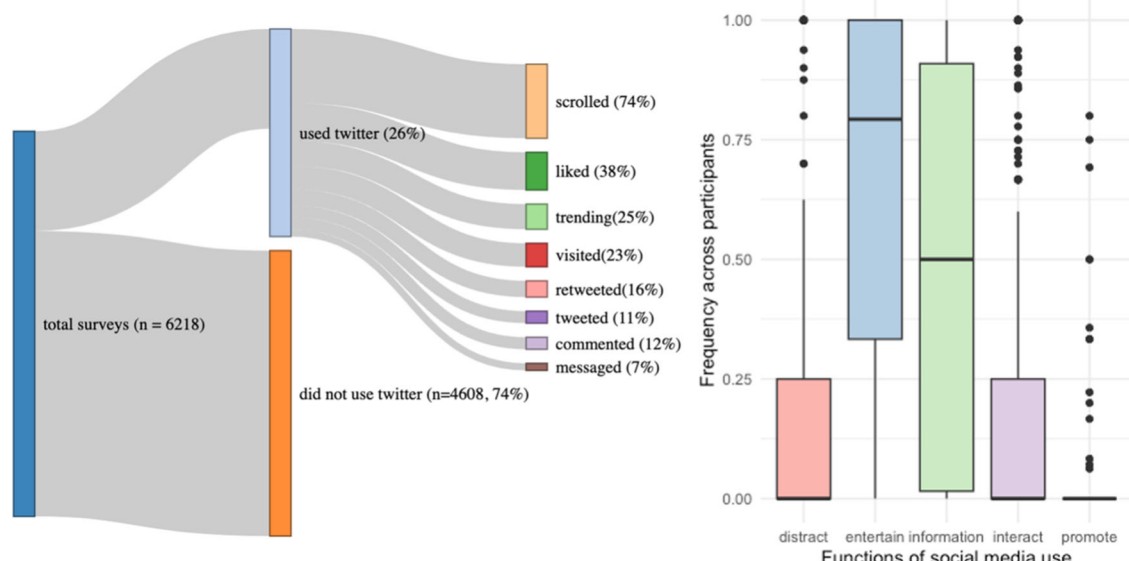

**Fig. 2 | Frequency of different types of Twitter use.** Left: Diagram of all the reported behaviors performed on Twitter. In dark blue, we have the total amount of surveys answered by the participants ($n$ = 6218). In the center, we have the proportion of the total answers in which participants reported Twitter use (26%) in light blue and the proportion of surveys in which participants did not report Twitter use (74%) in orange. On the right side of the lest panel, we have a breakdown of the frequency of each behavior on Twitter. Behaviors were not mutually exclusive, so they are not supposed to sum 100%. Note that the percentages of the behaviors are relative to the total amount of surveys in which they reported having used Twitter (e.g.: scrolling down was reported in 74% of 1610 surveys). Right: Boxplots of the percent of times that participants reported using Twitter for specific functions. The average was calculated across participants. All the percentages are relative to the total amount of times of surveys in which Twitter use was reported (1610). The Twitter functions were not mutually exclusive, so participants could select multiple in a single survey.

| | escapism | entertainment | self-promotion | information seeking | interacting |
|---|---|---|---|---|---|
| scrolled | 222 | 819 | 16 | 604 | 281 |
| liked | 142 | 443 | 24 | 325 | 213 |
| trending | 89 | 255 | 2 | 256 | 88 |
| visited | 93 | 279 | 18 | 232 | 144 |
| retweeted | 78 | 177 | 22 | 163 | 138 |
| tweeted | 59 | 143 | 21 | 86 | 106 |
| commented | 51 | 143 | 17 | 112 | 126 |
| messaged | 34 | 84 | 19 | 62 | 69 |

**Fig. 3 | Co-occurrence of the what and why of Twitter use.** A heat map of the frequency of reported behaviors (rows) and functions (columns). Darker colors indicate that the behavior and function co-occurred more often.

## Twitter use and polarization

Participants' current levels of affective polarization were operationalized as the absolute difference between warmth towards Democrats and Republicans. It was measured using a thermometer scale, where 0 was cold and 100 was warm[53]. Twitter use was significantly related to a 0.03 SD increase in polarization, $b_{within}$ = 0.03, CrI = [0.00, 0.06], at the within-person level, with a coefficient one-fifth of the absolute magnitude of our positive control benchmark, so quite small (Fig. 4).

We also asked participants if they had interacted with anyone who holds different values or a different worldview from theirs on Twitter ("yes" or "no"). We examined whether encountering people with different views or opinions would be related to changes in polarization but the credibility intervals included zero, $b_{within}$ = −0.05, 95% CrI = [−0.16, 0.74], $b_{between}$ = −4.52, 95% CrI [−0.13, 0.10], suggesting that the relationship might be too weak to be detected or non-existent.

## Twitter use and outrage

We conceptualized outrage as the sum of the emotions *angry, disgusted*, and *repulsed* (Cronbach's alpha of 0.80). Twitter use was related to a 0.19 SD increase in reported outrage at the within-person level, $b_{within}$ = 0.19,

CrI = [0.10, 0.28]. The magnitude of this effect was larger than the benchmark (Fig. 4). The large effect suggests that Twitter use and experiences of outrage tend to be closely related. We also tested if encountering people with different opinions would be related to changes in outrage. Our results did not suggest a relationship between the two variables, $b_{within}$ = 0.13, 95% CrI = [−0.53, 0.78], $b_{between}$ = 0.35, 95% CrI = [−0.31, 1.00], which could stem from low statistical power or a lack of association between the variables. A summary of the associations between Twitter use and the main outcome variables can be found in Table S3 in the supplementary information file.

## The what of Twitter use

Twitter behaviors were conceptualized as observable actions or activities they could do on Twitter, such as (1) scrolling down the feed, (2) liking tweets, (3) retweeting, (4) tweeting, (5) making a comment, (6) messaging, (7) seeing the trending topics, and (8) seeing others' profile. Participants indicated all the behaviors they performed in binary form. To examine *what aspects* of Twitter use are related to different consequences for the users, we ran models including all Twitter behaviors while controlling for past levels of the outcome variable. Samples for each behavior were restricted to the count of participants who reported engaging in them. We found that scrolling down the feed predicted a 0.08 SD (CrI = [−0.15, −0.01]) decrease in well-being at the within-person level with a magnitude half the size of our benchmark. Replying to others' tweets ($b_{within}$ = 0.23, CrI = [0.06, 0.4]), visiting trending topics ($b_{within}$ = 0.15, CrI = [0.05, 0.25]), and visiting others' profiles ($b_{within}$ = 0.15, CrI = [0.01, 0.28]) predicted increased sense of belonging at the within-subject level with effects at least as large as our benchmark. Retweeting behavior predicted 1.15 SD increases in polarization ($b_{within}$ = 1.15, CrI = [0.21, 2.15]) at the between-person level (i.e., people who tend to retweet a lot are more polarized). No Twitter behaviors were associated with changes in outrage. A summary of the relationships is reported in Fig. 4. For a full list of associations, see Table S4 and Fig. S2 in the Supporting Information File.

## The why of Twitter use

We conceptualized Twitter functions (the *why* of Twitter use) as the underlying motivations or reasons for the Twitter behaviors. For instance, I

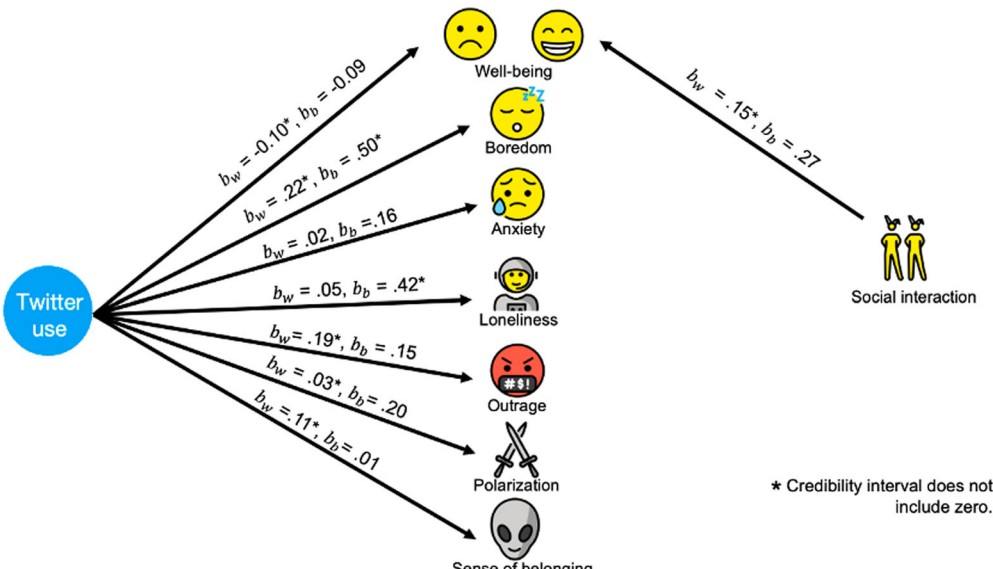

**Fig. 4 | Relationship between Twitter use and main outcome variables.** The diagram represents the relationship between Twitter use and the main outcome variables as well as the relationship between social interaction and well-being (our benchmark condition). $b_w$ is the coefficient for within-person differences, $b_b$ is the coefficient for between-person differences. Asterisks indicate that the credibility interval does not include zero.

can comment on a tweet because I want to socialize with others or because I want to relieve my boredom. Because there were no available measures for this construct, we consulted qualitative research on why people use social media[50] and then created and piloted a scale that captures functions of social media use into five factors: using social media for *entertainment, escapism, social interaction, self-promotion*, and *information seeking*. All details of the development of the *Functions of Social Media Use* scale can be found under supplementary methods in the Supporting Information file. In the surveys, participants answered "why did you use Twitter in the past 30 min?" by selecting one or more options from a list containing a description of the five factors. All responses were recorded in binary form. We ran models including all functions of use as predictors and controlling for past levels of the outcome. Using Twitter for *escapism* predicted decreases in well-being at both the within (0.25 SD, CrI = [−0.39, −0.12]) and between-person (1.43 SD, CrI = [−2.78, −0.25]) levels. Using Twitter for *entertainment* predicted a 0.04 within-person increase in polarization, CrI = [0.01, 0.08]. Using social media for *social interaction* predicted a 0.70 SD increase in between-person sense of belonging, CrI = [0.11, 1.31]. Finally, increased outrage was predicted by escapism at the between-person level (0.33 SD, CrI = [0.62, 2.9]) and information seeking at the within-person (0.14 SD, CrI = [0.03, 0.25]). Except for the effect on polarization, all effects were larger in magnitude than our benchmark. All relationships are reported in Fig. 5. These results suggest that different ways of using Twitter can effectively lead to varied outcomes at the within- and between-person levels.

**The who of Twitter use**
Even though we found aggregate effects of Twitter use on well-being, sense of belonging, polarization, and outrage, all effects were heterogeneous among participants. Figure 6 shows how much the coefficients (slopes) varied among participants. Note, for instance, that in the domain of well-being, despite the overall impact of Twitter use being negative, many participants experienced increased well-being with Twitter use.

Because of this high variability and research suggesting that the effects of social media use might be moderated by individual differences[24], we next probed the *who* of Twitter use, examining whether any individual variables, such as personality, age, and gender, would moderate the relationship between Twitter use and its various outcomes. We found no evidence that individual variables moderated the effects of Twitter. The lack of interactions could have resulted from our limited statistical power, so we compared

the model fit between models with and without interactions. The model fit comparison approach supported the null results for individual differences. All results are reported in Table 1. Although unexpected, this finding is corroborated by Johannes and colleagues[23], who found no evidence that reported social media use was associated with individual differences.

## Discussion
As the number of social media users increases every year, it is crucial to estimate how both people and society are affected by it. In the current study, we estimated the effects of Twitter use (now X), a social media platform that has the potential to shape public opinion and decision-making. Using the experience sampling technique over a period of one week, we found that Twitter use was related to overall within-person decreases in well-being and within-person increases in sense of belonging, polarization, outrage, and boredom. We also found that specific Twitter behaviors were linked to different outcomes: scrolling down the feed or using Twitter to distract oneself from problems (passive uses) were linked to within-person decreases in well-being, consistent with the idea that passively using social media is more detrimental to people[41,58]. Using it to connect with others on the platform (by replying to others, checking the trending topics, or checking other's profiles) was linked to within-person increases in sense of belonging; using it for entertainment was linked to within-person increases in polarization; while using it for information seeking was linked to within-person increases in outrage. All estimates were compared to a benchmark (the effect of a social interaction on well-being) and were comparable in magnitude: the smallest effect was one fifth of the benchmark while the largest was 25% larger.

The decomposition of variances into within-person and between-person components has shed light on specific associations unique to each level, potentially providing clarity to enduring debates in the literature. For example, a connection between loneliness and Twitter use emerged solely at the between-person level. This absence of association at the within-person level could imply that Twitter use does not inherently exacerbate loneliness. Instead, external factors contributing to loneliness might simultaneously drive individuals to use Twitter, although potential causal processes at extended timescales cannot be dismissed. Regarding other outcome variables, certain associations were discerned only at the within-person level, such as the impact of Twitter use on well-being, polarization, outrage, and sense of belonging. These findings may indicate that while fluctuations in

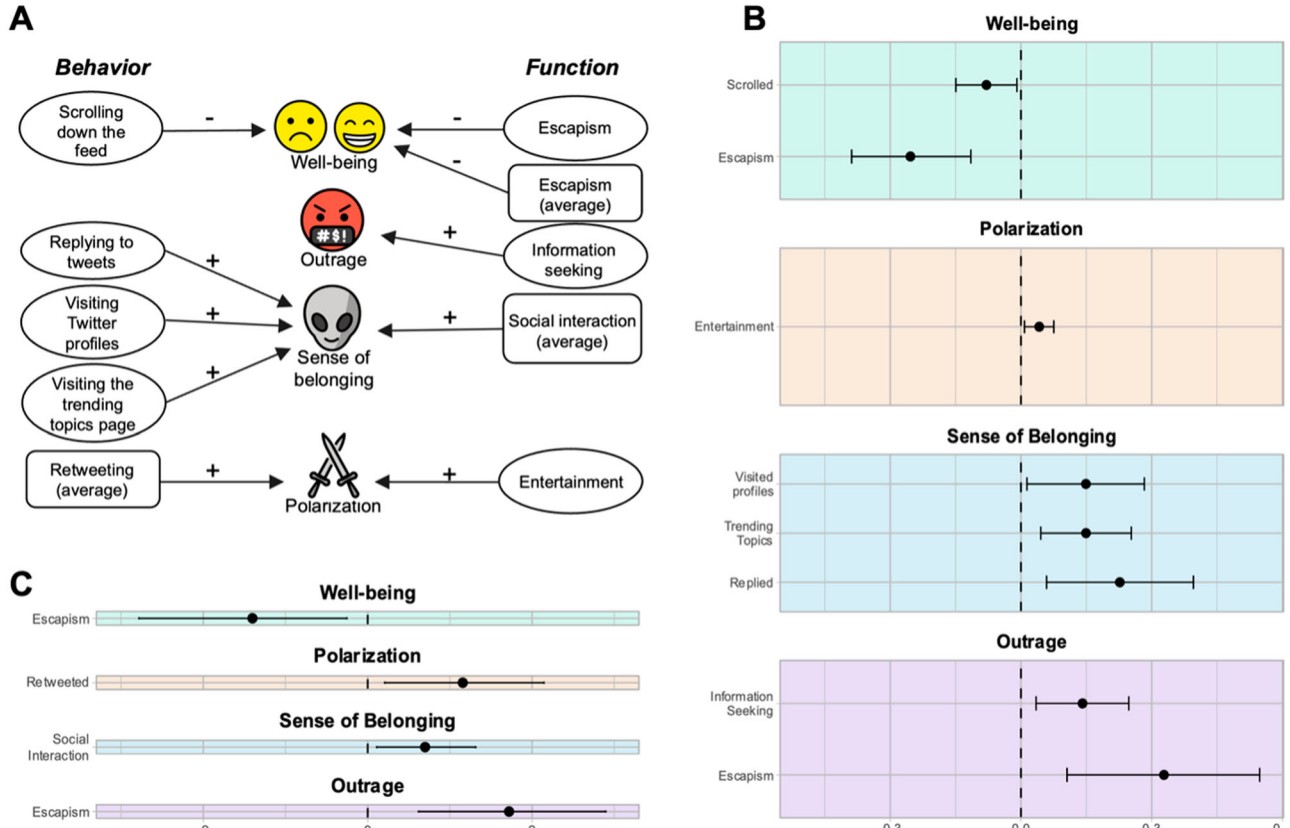

**Fig. 5 | Relationships between Twitter behaviors and functions with well-being, polarization, sense of belonging, and outrage. A** all the relationships between Twitter behaviors, functions of Twitter use, and the main outcome variables. Positive signs indicate a positive relationship. Oval boxes represent the effect at the within-person level, round rectangles are the relationships at the between-person level. **B** Z- scored coefficients for the types of use that predicted the outcome variables at the within-subject level. Bars represent the credibility interval. Dashed line represents zero. **C** Z-scored coefficients for the types of use that predicted the outcome variables at the between-subject level. Lines represent credibility interval ($n = 252$).

emotional states and Twitter use may coincide, these correlations do not persist over time to be observable in between-person analyses.

At the between-person level, people who used Twitter a lot were lonelier and more bored; people who retweeted a lot were more polarized; people who used Twitter to avoid their problems (escapism) had lower well-being and higher outrage levels; and people who used Twitter for social interactions had a higher sense of belonging. That people who retweet more often are more polarized is consistent with previous findings that most Twitter data are produced by a minority of users who tend to be more politically engaged than the average user[8]. The relationship between social interaction on Twitter and sense of belonging was also consistent with expectations: people who generally feel a stronger sense of community are more likely to do activities that involve social interactions. Finally—because avoidant coping strategies tend to be unsuccessful and cause emotional distress[59] and because well-being and outrage were operationalized as a function of negative emotional experiences—people who generally used Twitter to escape their problems probably felt less happy and more outraged due to the negative emotional consequences of using avoidant strategies to deal with their problems.

Why does whether someone uses Twitter or not relates to changes in psychological states? One explanation lies in *how* people use Twitter. When it comes to well-being, for instance, the predominance of passive or reactive types of use, such as scrolling down the feed, liking a tweet, and checking the trending topics, leaves users susceptible to the types of content they encounter on Twitter, which could lead users to engage in negative social comparisons and consume news, behaviors that are potentially linked to negative emotional consequences[24,60,61]. Our study design, however, did not

allow us to infer the direction of the relationships, so it is possible that what people feel influences their likelihood of using Twitter in specific ways.

Similarly, the relationship between Twitter use and increased outrage can be potentially explained by information seeking on Twitter. Since people are more likely to encounter moral violations on social media[5], they are more likely to get outraged when on social media, and the effect might be especially prominent when users go to the platform with the intention of looking for news or information. Although the effects of encountering moral violations seem especially prominent in the domain of politics, it is likely that it also occurs in non-political contexts[5,62]. Because effect direction cannot be determined, it is also possible that participants use Twitter when they are experiencing outrage. This finding also suggests that expressions of outrage on social media are not merely performative—as some might speculate—, as participants report feeling these emotions in surveys that only the researchers would see.

We also found that not only is Twitter associated with a higher sense of belonging, but that specific types of use, such as using Twitter for social interactions, replying to others' Tweets, visiting others' profiles, and checking the trending topics are especially linked to higher sense of belonging, corroborating our hypothesis that how people use Twitter matters. The first three types of use were not surprising: they refer to interactions with other people on the platform, which might generate a sense of community and friendship. The relationship between checking the trending topics and sense of belonging, however, is more puzzling. We speculate that because trending topics on Twitter contain news from topics of large cultural in-groups, such as politics or pop culture, users might have an increased sense of belonging while browsing them.

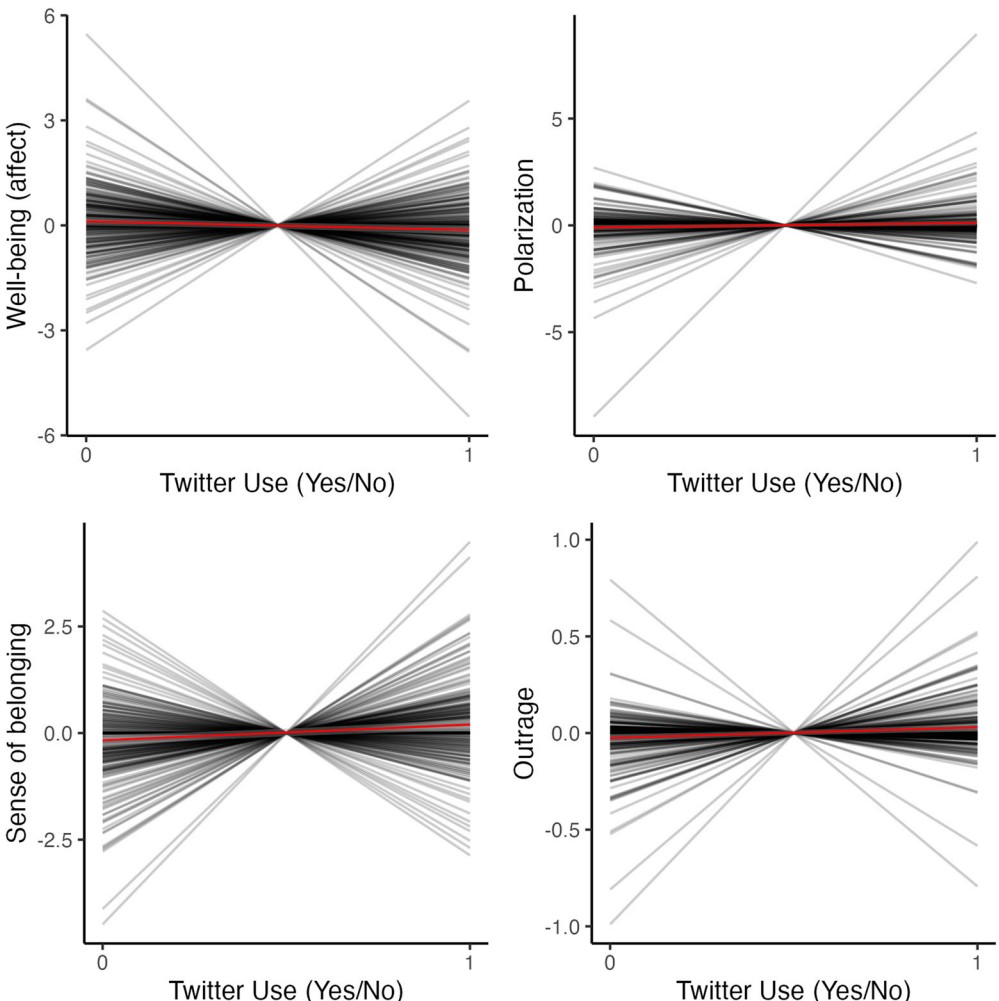

**Fig. 6 | Heterogeneity of the effects of Twitter use among participants for each outcome.** On each plot, we have a spaghetti plot where the *x* axis represents Twitter use (where zero is no twitter use and one is reported Twitter use), and the *y* axis is the centered outcome variable. In clockwise order, each plot represents the variability for well-being, polarization, outrage, and sense of belonging. Each grey line represents the regression slope for one participant when the outcome variable is mean centered around the participant mean. Red line represents the average slope. All models controlled for past levels of the DV and the IV. The outrage plot has 248 participants, the sense of belonging plot has 245 participants, the well-being plot has 246 participants, and the polarization plot has 248.

The relationship between Twitter use and polarization is more puzzling. One surprising finding was that using social media for entertainment was related to within-person increases in polarization. While an explanation is not obvious, we speculate about the possible directions of this relationship. It is possible that when participants experience events that increase polarization, they go to social media to clear their minds. This is consistent with the fact that most participants who said they went to Twitter for entertainment mostly scrolled down the feed (Fig. 5), supporting the idea that Twitter serves as a distraction. Another explanation is that people who use social media for entertainment may find satisfaction in engaging with divisive content. This might be similar to the way "trolls" enjoy provoking others online[63]. Still, it was not evident that *how* people use Twitter impacts their level of polarization. Instead, we suggest that more systemic influences, such as the existence of echo-chambers (in which users self-select to groups with similar worldviews[2,5,64]) contribute to increased polarization[65]. This hypothesis is also consistent with the fact that most public content is produced by a minority of users who are, on average, more polarized[34] and with our finding that polarized users retweet more often.

Taken together, these findings suggest that many of the effects of Twitter use are explained by psychological processes that happen within the same person and that they have practical significance in people's lives. There is an important caveat, however. Despite the aggregate effects having practical significance, they were heterogeneous among participants. This means, for instance, that a significant number of participants experienced increased well-being with increased Twitter use. Due to the high variability in the effects, we probed for individual differences that could moderate these effects but found no evidence that personality, age, or sex moderate these effects. It is possible that sex effects are more pronounced in samples of adolescents[66], so a more representative sample would not detect these effects. These findings suggest that individual factors might not be as relevant for social media as previously thought[24] or that individual differences other than the ones measured in our study moderate these relationships.

## Limitations

This study advances previous research on the psychological effects of Twitter use by employing a naturalistic intensive longitudinal design and a more representative sample. Although our results for the within-person analyses suggest that some processes happen at the intra-individual level, we cannot completely rule out the possibility of time-varying confounders (i.e., maybe when people are experiencing stressful events they both use Twitter more and have worse moods) or establish the direction of the intra-individual relationships (i.e., do I use Twitter when I am bored, or do I become bored when I am on Twitter?). Studies allowing for stronger causal inferences (e.g., experimental studies) are needed.

In trying to estimate precisely how much Twitter use influences psychological states over time, we ended up analyzing the relationship between

**Table 1 | Results of the interaction effects between Twitter use and the moderating variables**

| Moderator | Well-being | Polarization | Sense of belonging | Outrage |
|---|---|---|---|---|
| Neuroticism | b = 0.01, 95% CrI = [−0.08, 0.09], r = 0.02. ΔWAIC = −3.53, SE = 4.40. | b = 0.68, CrI = [−0.29, 1.65], r = 0.00. ΔWAIC = 6.23, SE = 5.69. | b = −0.00, CrI = [−0.07, 0.06], r = 0.02. ΔWAIC = 0.71, SE = 1.60. | b = 0.07, CrI = [−0.10, 0.23], r = 0.03. ΔWAIC = 12.55, SE = 16.01 |
| Agreeableness | b = −0.03, 95% CrI = [−0.19, 0.15], r = 0.02. ΔWAIC = −1.41, SE = 3.44. | b = −0.68, CrI = [−1.69, 0.34], r = 0.01. ΔWAIC = 13.29, SE = 5.67. | b = −0.02, CrI = [−0.09 0.04], r = 0.01. ΔWAIC = −1.06, SE = 2.69. | b = 0.05, CrI = [−0.17, 0.27], r = 0.01, ΔWAIC = 3.51, SE = 3.38. |
| Openness | b = 0.07, 95% CrI = [−0.04, 0.19], r = 0.01. ΔWAIC = 1.73, SE = 3.55. | b = 0.45, CrI = [−0.57, 1.44], r = 0.00. ΔWAIC = 19.06, SE = 9.40. | b = −0.03, CrI = [−0.10, 0.03], r = 0.02. ΔWAIC = −4.59, SE = 4.04. | b = 0.03, CrI = [−0.03, 0.09], r = 0.03. ΔWAIC = 2.61, SE = 4.97. |
| Conscientiousness | b = −0.02, 95% CrI = [−0.15, 0.10], r = 0.03.ΔWAIC = −10.39, SE = 6.46. | b = −0.83, CrI = [−1.86, 0.19], r = 0.01. ΔWAIC = 4.18, SE = 19.59. | b = 0.01, CrI = [−0.06, 0.07], r = 0.02. ΔWAIC = 2.75, SE = 1.21. | b = −0.00, CrI = [−0.07, 0.06], r = 0.03. ΔWAIC = 7.54, SE = 20.07 |
| Extraversion | b = −0.02, 95% CrI = [−0.13, 0.09], r = 0.02. ΔWAIC = −3.08, SE = 7.03. | b = −0.72, CrI = [−1.70, 0.28], r = 0.01. ΔWAIC = 3.84, SE = 6.32. | b = −0.02, CrI = [−0.09, 0.04], r = 0.02. ΔWAIC = −1.90, SE = 4.01. | b = 0.01, CrI = [−0.04, 0.07], r = 0.03. ΔWAIC = 11.77 SE = 7.57. |
| Age | b = −0.00, 95% CrI = [−0.01, 0.00], r = 0.01. ΔWAIC = 3.04, SE = 2.22. | b = 0.01, CrI = [−0.06, 0.09], r = 0.01. ΔWAIC = 2.64, SE = 19.32. | b = 0.00, CrI = [−0.00, 0.01], r = 0.01. ΔWAIC = 0.32, SE = 3.67. | b = 0.00, CrI = [−0.01, 0.01], r = 0.04. ΔWAIC = −3.55, SE = 20.44. |
| Gender | b = −0.01, 95% CrI = [−0.18, 0.17], r = 0.03. ΔWAIC = 1.35, SE = 2.52. | b = −0.96, CrI = [−2.93, 1.02], r = 0.01. ΔWAIC = 2.64, SE = 19.32. | b = 0.05, CrI = [−0.08, 0.17], r = 0.01. ΔWAIC = 2.15, SE = 3.61. | b = −0.20, CrI = [−0.45, 0.04], r = 0.02. ΔWAIC = −3.22, SE = 14.39. |

Each line corresponds to a moderating variable, and each column corresponds to a predictor. All models controlled for past levels of the DV and the IV. Differences in WAIC larger than 2 SDs show support for one of the models. Positive estimates suggest that the empty model was a better fit and negative estimates mean that the full model was a better fit.

variables that were a few hours apart. While understanding these close relationships is a first step to estimating the impact of Twitter, it is plausible that they do not extend to larger time periods (i.e., if I use Twitter for a month, how much does it affect me one year later?). While past studies have explored these effects on platforms such as Facebook[30], ongoing research should explore these relationships considering larger time windows on Twitter. Furthermore, future research should consider the potential impact of selection biases. The participants in our sample consisted exclusively of active Twitter users, which raises the possibility that those who experienced the most adverse effects from using Twitter may have already left the platform. Consequently, our findings might underestimate the negative influence of Twitter. Finally, while our study provides a comprehensive analysis of the effects of Twitter use, the comparative impact relative to other social media platforms remains unexplored. Consequently, we suggest systematic reviews of various platforms as a valuable avenue for future research.

## Conclusions
This study provides valuable insights into the effects of everyday Twitter (now X) use on well-being, emotions, sense of belonging, polarization, and outrage. Our findings demonstrate that a nuanced approach to the study of social media, for example, by categorizing the different affordances of the platform and dividing the effects between and within persons, allows for a better understanding of how different uses explain the diverse experiences of social media. Our findings suggest that Twitter use is associated with both positive and negative effects, including decreased well-being, increased sense of belonging, polarization, outrage, and boredom. Certain behaviors and motivations for Twitter use, such as scrolling down the feed and seeking information, contribute to these effects; however, we found no evidence that individual differences explain the variability in the effects. These findings emphasize the importance of considering both individual differences and momentary within-person changes over time when studying the psychological effects of Twitter use. They also indicate that future research ought to explore other potential moderating factors and long-term impacts to guide healthier social media engagement. Ultimately, understanding the complex relationship between social media use and psychological well-being will contribute to more informed decisions and healthier engagement with social media use.

## Data availability
The datasets used in this study are available at osf.io/e8krz.

## Code availability
The code is available at osf.io/e8krz.

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

## Acknowledgements
We would like to thank Hause Lin, Greg Depow, Zoe Francis, Amanda Ferguson, Taylor Sparrow-Mungal, and Artur Back de Luca for their help. This research was made possible by generous support from an Insight Development Grant from the Social Sciences and Humanities Research Council of Canada to M.I. The funders had no role in study design, data collection and analysis, decision to publish or preparation of the manuscript.

## Author contributions
V.O.M. conducted the study. M.I. & F.C. jointly supervised this work.

## Competing interests
The authors declare no competing interests.
