## [Peer Review File · Communications Psychology]

This manuscript has been previously reviewed at another Nature Portfolio journal. This document only contains reviewer comments and rebuttal letters for versions considered at Communications Psychology

5th Dec 23

Dear Ms Oldemburgo de Mello,

Your manuscript titled "Twitter use predicts substantial changes in well-being, polarization, sense of belonging, and outrage" has now been seen by our reviewers, whose comments appear below. Both reviewers had previously commented on your work before you transferred it to us (numbering as previously).

In light of their advice I am delighted to say that we are happy, in principle, to publish a suitably revised version in Communications Psychology under the open access CC BY license (Creative Commons Attribution v4.0 International License).

We therefore invite you to revise your paper one last time to address the remaining concerns of our reviewers and a list of editorial requests. At the same time we ask that you edit your manuscript to comply with our format requirements and to maximise the accessibility and therefore the impact of your work.

EDITORIAL REQUESTS:

Beyond the individual instances highlighted in the table, we ask you to revise the work with the following three key issues in mind:

First, the work does not demonstrate causality and causal language needs to be avoided. The manuscript is largely compliant with this guideline, but please ensure that you refer to the temporal associations, rather than invoking causality throughout.

Second, the changes you measure are short-term and the timescale of measurement needs to be made clear throughout, including in the Abstract.

Finally, you obtained data from Twitter, which subsequently changed its name to X. Where you refer to features of the platform based on past references, it's appropriate to use "Twitter". Where you describe the data and results, it's likewise appropriate to refer to "Twitter". However, please do mention early on that the platform has now changed its name (so that readers in the future can make the link) and ensure that potential statements about future research use the correct (new) name.

SUBMISSION INFORMATION:

In order to accept your paper, we require the files listed at the end of the Editorial Requests Table; the list of required files is also available at <https://www.nature.com/documents/commsj-file->

checklist.pdf .

OPEN ACCESS:

Communications Psychology is a fully open access journal. Articles are made freely accessible on publication under a [CC BY](http://creativecommons.org/licenses/by/4.0) license (Creative Commons Attribution 4.0 International License). This license allows maximum dissemination and re-use of open access materials and is preferred by many research funding bodies.

For further information about article processing charges, open access funding, and advice and support from Nature Research, please visit <https://www.nature.com/commspsychol/article-processing-charges>

At acceptance, you will be provided with instructions for completing this CC BY license on behalf of all authors. This grants us the necessary permissions to publish your paper. Additionally, you will be asked to declare that all required third party permissions have been obtained, and to provide billing information in order to pay the article-processing charge (APC).

* **DATA AVAILABILITY:**

[link redacted]

** This url links to your confidential home page and associated information about manuscripts you may have submitted or be reviewing for us. If you wish to forward this email to co-authors, please

delete the link to your homepage first **

Best regards,

Marike

Marike Schiffer, PhD
Chief Editor
Communications Psychology

REVIEWERS' COMMENTS:

Reviewer #1 (Remarks to the Author):

I appreciate the authors' thoughtful replies to my comments and their softening of the causal language. I think the paper is much improved, and I have no further concerns.

Reviewer #3 (Remarks to the Author):

The authors have appropriately addressed all of my comments and I am happy to recommend publication.

Thank you for the impactful work.